# Colorectal Cancer Patients Have Four Specific Bacterial Species in Oral and Gut Microbiota in Common—A Metagenomic Comparison with Healthy Subjects

**DOI:** 10.3390/cancers13133332

**Published:** 2021-07-02

**Authors:** Yoshinori Uchino, Yuichi Goto, Yusuke Konishi, Kan Tanabe, Hiroko Toda, Masumi Wada, Yoshiaki Kita, Mahiro Beppu, Shinichiro Mori, Hiroshi Hijioka, Takao Otsuka, Shoji Natsugoe, Eiji Hara, Tsuyoshi Sugiura

**Affiliations:** 1Department of Maxillofacial Diagnostic and Surgical Science, Field of Oral and Maxillofacial Rehabilitation, Graduate School of Medical and Dental Sciences, Kagoshima University, 8-35-1, Sakuragaoka, Kagoshima 890-8544, Japan; k2309975@kadai.jp (Y.U.); ygoto@dent.kagoshima-u.ac.jp (Y.G.); mbeppu@dent.kagoshima-u.ac.jp (M.B.); zio@dent.kagoshima-u.ac.jp (H.H.); 2Department of Molecular Microbiology, Research Institute for Microbial Diseases, Osaka University, 3-1, Yamadaoka, Suita, Osaka 565-0871, Japan; ykonishi@biken.osaka-u.ac.jp (Y.K.); ehara@biken.osaka-u.ac.jp (E.H.); 3Department of Digestive Surgery, Breast and Thyroid Surgery, Graduate School of Medical Sciences, Kagoshima University, 8-35-1, Sakuragaoka, Kagoshima 890-8520, Japan; k3113670@kadai.jp (K.T.); north-y@m.kufm.kagoshima-u.ac.jp (Y.K.); morishin@m3.kufm.kagoshima-u.ac.jp (S.M.); takao-o@kufm.kagoshima-u.ac.jp (T.O.); 4Breast Surgery, Fujita Health University Hospital, 1-98, Dengakubo, Kutsukake, Toyoake, Aichi 470-1192, Japan; hiroko.toda@fujita-hu.ac.jp; 5Department of Digestive Surgery, Imakiire General Hospital, 43-25, Korai, Kagoshima 890-0051, Japan; m.wada3373@imakiire.or.jp; 6Kajikionsen Hospital, 4714, Kida, Kajiki, Aira, Kagoshima 899-5241, Japan; s-natsugoe@gyokushoukai.com

**Keywords:** microbiota, oral bacteria, colorectal cancer, *Peptostreptococcus*, *Streptococcus*, *Solobacterium* spp.

## Abstract

**Simple Summary:**

The incidence of colorectal cancer (CRC) has been increasing in recent years, and the gut microbiota is nowadays considered to be involved in the progression of CRC. Recent studies have investigated the involvement of the oral microbiota in CRC development using saliva and stool samples. However, the details regarding how oral bacteria alter the gut microbiota and affect CRC carcinogenesis remain unclear. In the present study, we identified four bacterial species that may affect the carcinogenesis and progression of CRC. These microorganisms may be potential biomarkers in saliva for diagnosing CRC.

**Abstract:**

Oral microbiota is reportedly associated with gut microbiota and influences colorectal cancer (CRC) progression; however, the details remain unclear. This study aimed to evaluate the role of oral microbiota in CRC progression. Fifty-two patients with CRC and 51 healthy controls were included. Saliva and stool samples were collected, and microbiota were evaluated using 16S rRNA analysis and next-generation sequencing. Comparative analysis was performed on both groups. Linear discriminant analysis effect size (LEfSe) revealed the presence of indigenous oral bacteria, such as *Peptostreptococcus*, *Streptococcus*, and *Solobacterium* spp., at a significantly higher relative abundance in saliva and stool samples of CRC patients compared with controls. Next, CRC patients were divided into early stage (Stage I, II; *n* = 26; 50%) and advanced stage (Stage III, IV; *n* = 26; 50%) disease. LEfSe revealed that *S. moorei* was present at a significantly higher relative abundance in the advanced-stage group compared with the early-stage group, again consistent for both saliva and stool samples. Among bacterial species with significantly higher relative abundance in CRC patients, *P. stomatis*, *S. anginosus*, *S. koreensis*, and *S. moorei* originated from the oral cavity, suggesting indigenous oral bacteria may have promoted initiation of CRC carcinogenesis. Furthermore, *S. moorei* may influence CRC progression.

## 1. Introduction

The number of patients with colorectal cancer (CRC) has markedly increased in recent years [1]. In 2002, the number of new CRC patients worldwide was estimated to be approximately 1.02 million but exceeded 1.8 million in 2018. Deaths were estimated at 881,000, representing approximately 1 death for every 10 cancer cases. Overall, CRC ranked third in incidence and second in mortality [2]. Statistics from the United States showed that the number of new CRC cases significantly increased in 2014 in both men and women over the age of 50. CRC mortality rate has been decreasing since 1975, which has been attributed to increased screening [3] considering that more countries have established CRC screening for people over the age of 40 or 50. Screening is expected to result in early detection and treatment of CRC, leading to a subsequent reduction in mortality. The international strategy is to suppress the development of CRC.

Oral microbiota is considered to play an important role as a reservoir for intestinal microbiota and bacterial infections, but the details remain unclear. Hara and colleagues demonstrated the importance of gut microbiota as a contributor to liver cancer [4,5]. Their most recent research indicated that approximately half of the bacterial species thought to contribute to CRC are oral bacteria. Indeed, it has been reported that saliva samples of patients with CRC and CRC tissue show the same strain of *Fusobacterium nucleatum*, suggesting that the *F. nucleatum* of CRC originates in the oral cavity [6]. Yu et al. have reported that oral bacteria, such as *F. nucleatum* and *Parvimonas micra*, are consistently enriched in stool samples of patients with CRC compared to those of healthy individuals. These investigators also reported that the abundance of these two species of bacteria is significantly higher in samples from patients with stage II or higher CRC compared to control samples [7]. Based on these findings, it is possible that oral microbiota is related to gut microbiota and has some influence on the progression of CRC; however, the details are still unknown.

In the present study, we investigated the role of the oral cavity as a reservoir for gut microbiota and evaluated the possible involvement of oral microbiota in CRC progression. We conducted 16S rRNA gene analysis of the microbiota in saliva and stool samples collected from patients with CRC and healthy adults. A comparative analysis of the saliva and stool microbiota of both groups was then performed.

## 2. Materials and Methods

### 2.1. Sample Collection

In this study, 56 patients diagnosed with CRC at the Digestive Surgery Department of Kagoshima University Hospital, Kagoshima, Japan (disease group: D) and 51 healthy volunteers aged ≥40 years (control group: N) were included. Saliva and stool samples were collected, and individual microbiota evaluated using 16S rRNA analysis and next-generation sequencing. A comparative analysis of both groups was also performed. The study was conducted in accordance with the Declaration of Helsinki. The study protocols were approved by the ethics committee of the Kagoshima University. Informed consent was obtained from all the study participants.

The saliva and stool samples were self-collected by the participants after waking up in the morning and prior to eating, drinking, gargling, or teeth brushing. Both samples were collected on the same day. Patients with CRC were sampled before the start of treatment (surgery or neoadjuvant chemotherapy). Samples from healthy controls who had no oral or gastrointestinal malignant disease were included. Collected samples were excluded for the following cases: (i) antibiotic use within 1 week, (ii) constipation or diarrhea with Bristol Stool Form Scale scores ≤5 sampled, and (iii) consumption of alcohol the previous day. Saliva samples were collected using the OMNIgene-ORAL OM-501 Saliva Microbiome DNA Collection Kit (DNA Genotek, Inc., Ottawa, ON, Canada) and stool samples were collected using the Stool Collection Kit FS-0006 (Techno Suruga Laboratory Co., Shizuoka, Japan), according to manufacturers’ instructions. DNA extraction was performed at Biken Biomics, Inc. (Osaka, Japan). All collected samples were immediately stored below 4 °C until delivery to the laboratory.

### 2.2. DNA Preparation and Microbiota Analysis

Genomic DNA was extracted using a GENE STAR PI-480 automated DNA isolation system (Kurabo Industries, Ltd., Osaka, Japan). After extraction, the amount of DNA in the samples was measured using the Qubit^TM^ dsDNA HS assay kit (Thermo Fisher Scientific, Waltham, MA, USA). Polymerase chain reaction (PCR) analysis was performed using primers 16S-27Fmod (5′-TCGTCGGCAGCGTCAGATGTGTATAAGAGACAGAGRGT TTGATYMTGGCTCAG-3′) and 16S-338R (5′-GTCTCGTGGGCTCGGAGATGTGTATA AGAGACAGTGCTGCCTCCCGTAGGAGT-3′) for the V1-V2 region of bacterial 16S rRNA and the KAPA HiFi Hot Start Ready Mix (Roche, Basel, Switzerland). PCR reactions were performed based on the 16S library preparation protocol (Illumina Inc., San Diego, CA, USA). The size of the mixed libraries was confirmed by agarose gel electrophoresis and the concentration was quantified by real-time PCR. Paired-end sequencing (250 bp) was performed using a MiSeq Reagent Kit v2 (500 cycles). The acquired sequence data were analyzed using Qiime2-2019.10 software [8]. Paired-end reads were merged, sequence errors removed, and sequence clustering performed. Operational taxonomic unit (OTU) clustering with a threshold of 97% was applied in our study. For taxonomic analysis, each OTU was aligned with q2-feature-classifier and the Greengenes 13_8 99% OTUs reference database.

### 2.3. Statistical Analysis

Differences between groups for participants’ characteristics were analyzed using the *χ*^2^ or Welch’s *t* test. Linear discriminant analysis Effect Size (LEfSe) [9] was used to determine the relative abundance of significantly different species between the CRC patients and controls or between early-stage and advanced-stage CRC. Tables containing the relative abundance of the species were imported into LEfSe (ver. 1.0) on the web-based Galaxy (http://huttenhower.org/galaxy/) (accessed on 10 June 2020) server, and logarithmic linear discriminant analysis (LDA) scores were calculated online. The content of the OTU table was aligned to the 16S ribosomal RNA sequences in the National Center for Biotechnology Information (NCBI) database by performing a BLAST search (https://blast.ncbi.nlm.nih.gov/Blast.cgi) (accessed on 10 June 2020). Scatter plots were created using the package “ggplot2” of the R program.

## 3. Results

### 3.1. Patient Characteristics

Four group D patients were excluded from analysis after DNA extraction due to insufficient DNA and no confirmed diagnosis of “adenocarcinoma”. Therefore, a total of 206 samples were ultimately analyzed (103 saliva and 103 stool). The clinical characteristics of the participants in each group are shown in Table 1. The 51 control group N participants were 54.49 ± 10.6 years of age, and the 52 group D patients were 68.52 ± 10.6 years of age. Group D did not differ from group N in terms of sex ratio. Gingival plaque was richer in group D than that in group N (*p* < 0.01). The number of teeth brushing times (per day) in group D was lower than that in group N (*p* < 0.01), suggesting poor oral hygiene in the patients with CRC. No significant differences were observed with regards to alcohol consumption or smoking (Table 1). The clinicopathological characteristics of the patients with CRC are shown in Table 2. Early-stage disease (Stage I, II) and advanced-stage disease (Stage III, IV) were found to be equally distributed with 26 of the patients with CRC being in each group.

### 3.2. Saliva and Stool Microbiota Differences between CRC Patients and Controls

We analyzed the microbiota in saliva and stool samples from the 52 patients with CRC and 51 controls. The relative abundance of microbiota makeup at the class level in each of the four groups is shown in Figure 1. The microbiota composition was significantly different between the saliva and the stool samples, indicating a high habitat specificity for the sample type.

The diversity analysis results are shown in Figure 2. Permutational multivariant analysis of variance (PERMANOVA) of the weighted UniFrac distance analysis was performed (Figure 2a). The weighted UniFrac distance results were not significantly different between the saliva samples of the controls and patients with CRC (PERMANOVA, *p* = 0.07), but was significantly different for the stool samples (PERMANOVA, *p* = 0.001). Principal coordinate analysis (PCoA) plots based on weighted UniFrac distances of the microbiota in each sample are shown in Figure 2b. Consistent with the results shown in Figure 1, the PCoA plots indicated the saliva and stool samples formed clearly separate groups but with similar microbiota compositions.

### 3.3. Potential Biomarker Bacterial Species Based on Lefse Analysis

The tables containing the relative abundance of the microbial species in the study samples were imported into LEfSe on the web-based platform Galaxy. The LEfSe analysis results comparing the control group N and group D patients with CRC are shown in Figure 3. Comparison results of the saliva samples are shown in Figure 3a and those for the stool samples are shown in Figure 3b. Bacterial species with a green label had higher LDA scores in the patients with CRC compared to that in the controls, indicating significantly higher relative abundance of those bacterial species in the patients with CRC compared to that in the controls. In contrast, the bacterial species with a red label had higher LDA scores in the controls compared to that in the patients with CRC, indicating significantly higher relative abundance of those bacterial species in the controls. Four bacterial species were identified that had a significantly higher relative abundance in both saliva and stool of CRC patients (group D) compared to those in the control group N. The four indigenous oral bacteria strains were *Peptostreptococcus stomatis* strain W2278, *Streptococcus anginosus* SK52 = DSM 20563, *Solobacterium moorei* strain JCM 10645, and *Streptococcus koreensis* strain KCOM 2890. Scatter plots of the results for the four bacterial species were prepared using the package “ggplot2” of the R program (Figure 3c).

The CRC patients (group D) were then divided into early-stage disease (Stage I, II; *n* = 26; 50%) and advanced-stage disease (Stage III, IV; *n* = 26; 50%) and LEfSe analysis performed on the subgroups. Results for the saliva of CRC patients (D-saliva) are shown in Figure 4a and the results for the stool of CRC patients (D-stool) are shown in Figure 4b. Considering the results shown in Figure 3a,b, bacterial species with green labels had higher LDA scores among the “advanced-stage” group compared to that among the “early-stage” group, indicating that those bacterial species had significantly higher relative abundance in advanced-stage of disease. In contrast, the bacterial species with red labels had higher LDA scores among the “early-stage” group compared to that among the “advanced-stage” group, indicating those bacterial species had significantly higher relative abundance was significantly higher in early-stage of disease. Of the four bacterial species with high relative abundance in the patients with CRC shown in Figure 3, *S. moorei* strain JCM 10645 had a significantly higher relative abundance in the advanced-stage group compared to that in the early-stage group for both saliva and stool. Scatter plots of the relative abundance of *S. moorei* strain JCM 10645 in saliva and stool samples according to the degree of CRC progression were created (Figure 4c).

## 4. Discussion

In the present study, we performed a comparative analysis of the oral and intestinal microbiota of patients with CRC and healthy individuals and revealed the composition of the microbiota of each group of samples at the class level. The class of bacteria with the highest relative abundance in saliva samples of healthy subjects and patients with CRC was Bacilli. In comparison, the class of bacteria with the highest relative abundance in stool samples was *Clostridia.* Previous reports have shown that bacterial species composition differs greatly between the oral cavity and gastrointestinal tract and habitat specificity is high [10,11]. Similarly, the results of our current study showed that saliva and stool samples had different microbiota compositions.

Weighted UniFrac distance analysis revealed a significant difference between patients with CRC and controls in stool samples, but not in saliva samples. This suggests there was a significant difference in β-diversity of the intestinal microbiota between controls and patients with CRC, but not in β-diversity of the oral microbiota. This result is consistent with the result of Liu W et al. in that the diversity of intestinal microbiota showed a significant separation between patients with CRC and healthy individuals [12]. It has been reported that saliva is richer in bacterial microbiota than other oral components, such as the buccal mucosa and keratinized gingiva [13]. However, Huse et al. reported that β diversity in the oral bacterial microbiota is low compared to that in other parts of the body [14], which is in agreement with our study results.

LEfSe analysis using the Galaxy platform showed that the relative abundances of *P. stomatis*, *S. anginosus*, *S. moorei*, and *S. koreensis* were higher in patients with CRC compared to those in controls, in both saliva and stool samples. All these microorganisms are normal microbiota of the oral cavity with the above four bacterial species being supplied to the large intestine from the oral cavity, suggesting they may contribute to the carcinogenesis of CRC. The involvement of *F. nucleatum* in colorectal carcinogenesis has been noted in many papers [15,16,17,18,19]. Invasion of bacterial species and metabolites into the tumor microenvironment enhances tumor growth by inducing an immune cell response that promotes tumors [20]. In particular, repression of the *F. nucleatum* tumor suppressor gene adenomatous polyposis coli (Apc) accelerated tumorigenesis in the small intestine and colon of gene-mutated mice (Apc^Min/+^ mice) [21]. Metabolites of *F. nucleatum* directly promote tumor cell proliferation and immune cell infiltration, making the tumor microenvironment more tumor-tolerant over time. It has also been shown that *F. nucleatum* attaches and invades endothelial and epithelial cells, induces carcinogenic and inflammatory responses, and stimulates CRC cell growth through FadA adhesin [22]. However, details regarding the route used by *F. nucleatum* from the oral cavity to the intestine and its effect on carcinogenesis are largely unknown. Komiya et al. detected the same strain of *F. nucleatum* in both colon cancer tissue and saliva of patients with CRC using arbitrary primer (AP)-PCR. This indicates that *F. nucleatum*, which is frequently detected in CRC tissues and suspected to be involved in carcinogenesis, is derived from the oral cavity [6]. In the present study, *F. nucleatum* subsp. *nucleatum* (similarity 98.563% in BLAST search OTU) had a significantly higher relative abundance in only the saliva of patients with CRC. In addition, *F. nucleatum* subsp. *vincentii* (similarity 99.424% in BLAST search OTU) had a higher relative abundance in only the stool of patients with CRC. These bacteria are both *F. nucleatum* but differ at the subspecies and strain level. When each patient sample was examined for the presence or absence of each bacterial species, *F. nucleatum* subsp. *nucleatum* was detected in the saliva of 11/52 (21.15%) patients with CRC, but not detected in the stool of these patients. In contrast, *F. nucleatum* subsp. *vincentii* was detected in the saliva samples of 35/52 (67.3%) patients with CRC, as well as in the stool of 6 of these patients. These results are consistent with the results of Komiya et al. in that a common strain of *F. nucleatum* subsp. *nucleatum* is not detected in both CRC tissue and saliva, but *F. nucleatum* subsp. *vincentii* is observed as a common strain in both samples [6]. According to these investigators, *F. nucleatum* subsp. *vincentii* may be sourced to the intestine or colonized in tissues as an influx from the oral cavity. However, *F. nucleatum* subsp. *nucleatum* can colonize tissues, but may not be derived from the oral cavity. According to the results of the previous report [6] and our current study, we hypothesize that *F. nucleatum* subsp. *nucleatum* colonizes tissue of the large intestine, but its supply from the oral cavity to the intestine is difficult and it has little effect on CRC progression. However, it is possible that *F. nucleatum* subsp. *vincentii* is constantly supplied from the oral cavity and settles in intestine tissues, affecting CRC progression.

Most previous studies have reported that *F. nucleatum* is derived from the oral cavity and affects CRC progression. However, our findings showed that *F. nucleatum* may or may not be derived from the oral cavity, depending on the subspecies and strain, and some may or may not affect CRC progression. Furthermore, oral hygiene tended to be worse in patients with CRC compared to that in controls in previous studies [23,24,25]. Similarly, we found poor oral hygiene in our cohort of patients with CRC. Therefore, the number of bacterial species noted above may be reduced by intervention with active oral cleaning, which may reduce the influx of bacterial species into the intestine, thereby leading to the prevention of CRC.

Our results also showed that the relative abundance of *S. moorei* was higher in both saliva samples and stool samples of CRC patients with stage III and IV disease compared to that in patients with stages I and II disease. This suggests that *S. moorei* may affect not only CRC carcinogenesis but also CRC progression. Yu et al. showed that the relative abundances of *P. micra*, *F. nucleatum*, *S. moorei*, and *P. stomatis* are significantly higher in fecal samples of patients with CRC than those in healthy subjects [7]. Similar results are observed for the above species regarding their high relative abundance in stool samples. In our study, *S. moorei* and *P. stomatis* were present at relatively higher amounts, not only in the stool but also in the saliva. This finding suggests that these two species in the intestine may be derived from the oral cavity. In the study by Yu et al., *F. nucleatum* and *P. micra* showed significantly higher abundances in feces of CRC patients with stage II and stage III disease compared to those in feces of patients with stage I disease and healthy controls. Despite previous reports on the relation between CRC and the abovementioned bacterial species, including *S. moorei* identified in our study, the specific mechanism affecting CRC progression has not yet been elucidated. *S. moorei* can be isolated from feces and the oral cavity and is a bacterial species mainly associated with endodontic infection and periodontal disease. Based on a case report, the clinical characteristics of patients with *S. moorei* bacteremia are associated with debilitating conditions such as malignancies and intravenous drug use across gender and age [26]. It is expected that the pathological condition of *S. moorei* is a factor that influences progression of existing CRC, via creating an inflammatory environment. Investigating the precise cause and mechanism of *S. moorei* and CRC progression will be a topic for future study. The genus Streptococcus, including *S. koreensis* and *S. anginosus*, are also generally considered commensal bacteria of the human oral cavity and can be isolated from the subgingival dental plaque of periodontitis lesions [27,28]. *S. anginosus* is also recognized as belonging to the normal flora of the human gastrointestinal tract, and there are case reports in which colorectal cancer was found in patients *S. anginosus* bacteremia [28]. Further research is needed to determine whether *S. anginosus* infection is a risk factor for CRC or a consequence from cancerous lesion-derived insult to the normal mucosa allowing pathogens to invade the host circulation.

In this study, we focused on the base sequences of bacterial species using next-generation sequencing for analysis. Bacterial species were identified using the sequence data of a limited region of 16S rRNA. However, future studies should perform whole genome shotgun sequencing to obtain information on the entire bacterial genome to confirm the existence of the target bacterial species. Moreover, further studies that characterize the four bacterial species identified in this study will be needed to elucidate their role in CRC. To validate the present biomarker study, a screening PCR system should be established to confirm the existence of the target bacterial species in saliva, and a large cohort study should be performed. In this study, the average age of the controls and patients with CRC differed significantly (*p* < 0.01). Although the gut microbiota reportedly changes with age [29,30,31,32], our study focused on the supply of oral bacteria to the gut microbiota. As a result of narrowing down the conditions for those aged 40 years or older without oral malignant disease and digestive malignant disease, the sample was as shown in this case. Regarding the sample size of this study, we performed post hoc analysis using G power application [33,34]. When we used the relative ratio of each bacterial strain as CRC detecting index, the powers were around 0.85 for *P. stomatis* and *S. moorei*. So we confirmed that a sample size of 52:51 would be sufficient for these two strains. However, more samples are necessary for another two strains. Hence, we conclude that the sample size used in this study is acceptable as a comparison study. Through LEfSe analysis, we aimed to identify potential strains for use as biomarkers based on variations in bacterial microbiota due to differences in eating habits; however, no definitive result was obtained. This is likely due to the limited number of samples. Previous studies have suggested that dietary changes affect the composition of intestinal microbiota, which is associated with fragility, nutritional status, and various diseases [35,36,37,38,39,40]. As such, the effect of dietary changes on the intestinal microbiota in regard to CRC carcinogenesis is also a topic for future study.

## 5. Conclusions

In the present study, we identified alteration of four bacterial species in saliva and gut microbiota with CRC that suggest the possibility of these organisms play some role in the carcinogenesis and progression of CRC. These four strains also have the potential to be used as biomarkers in saliva for diagnosing CRC. In further studies, it will be necessary to increase the number of samples and perform experiments using PCR analysis.

## Figures and Tables

**Figure 1 cancers-13-03332-f001:**
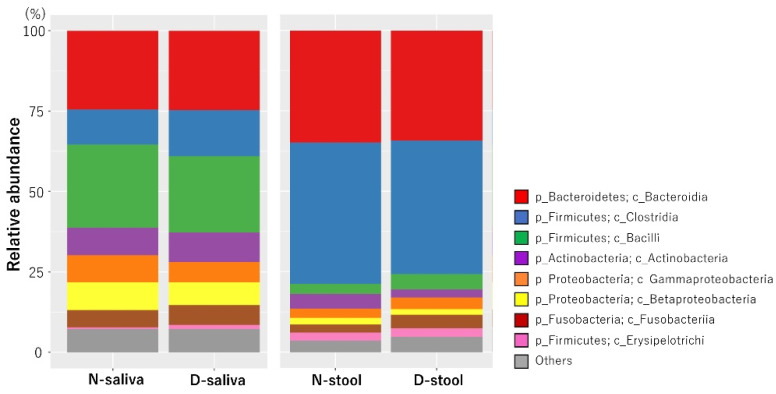
Microbiota composition at the class level for each sample group. Each label represents the average relative abundance. The most abundant class of bacteria found in saliva samples was Bacilli. The most abundant class of bacteria found in stool samples was Clostridia. N, control; D, patients with CRC; CRC, colorectal cancer.

**Figure 2 cancers-13-03332-f002:**
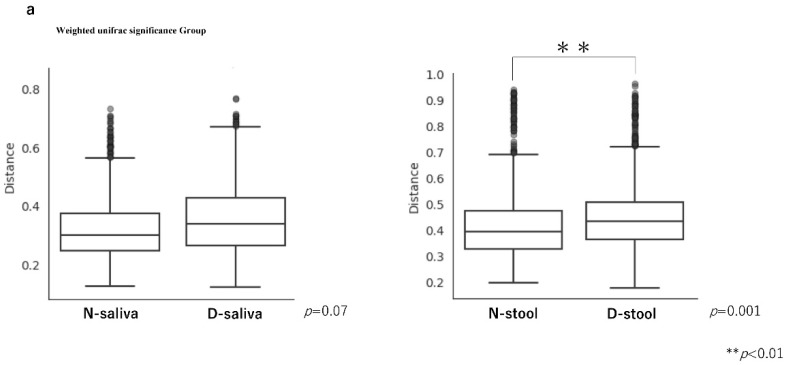
Diversity analysis. (**a**) Although β diversity was not significantly different in saliva samples between controls and patients with CRC, it was significantly different in stool samples between the groups. The significance test was evaluated using permutational multivariant analysis of variance (PERMANOVA). (**b**) Principal coordinate analysis (PCoA) plots based on weighted UniFrac distance analysis of the microbiota in each sample. Closer plots indicate a more similar microbiota composition. Axis 1 and Axis 2 indicate the percentage of variation explained by principal coordinates.

**Figure 3 cancers-13-03332-f003:**
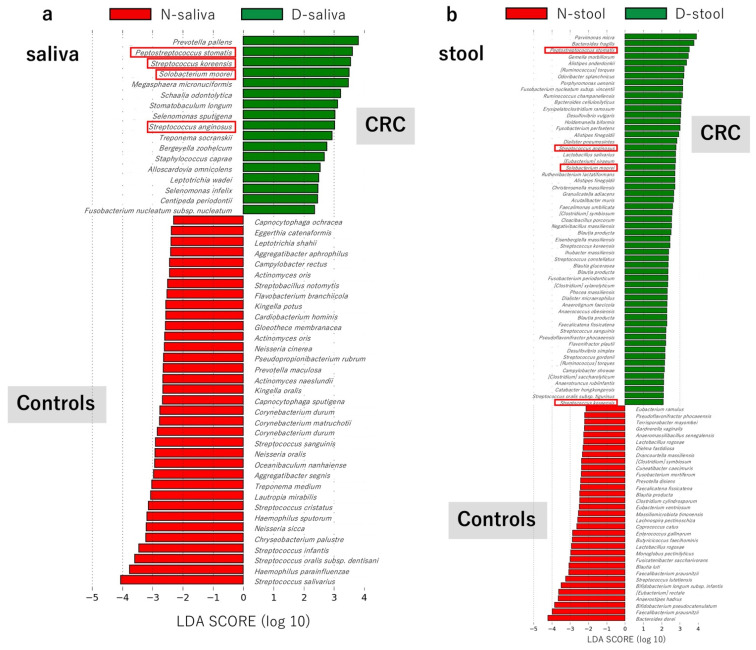
Linear discriminant analysis effect size (LEfSe) comparing bacterial group abundances between controls (group N) and patients with CRC (group D) in saliva (**a**) and stool (**b**) samples. (**c**) The relative abundance of four indigenous oral bacteria, *Peptostreptococcus stomatis*, *Streptococcus anginosus*, *Solobacterium moorei,* and *Streptococcus koreensis*, was statistically significant in saliva and stool of patients with CRC compared to those in controls.

**Figure 4 cancers-13-03332-f004:**
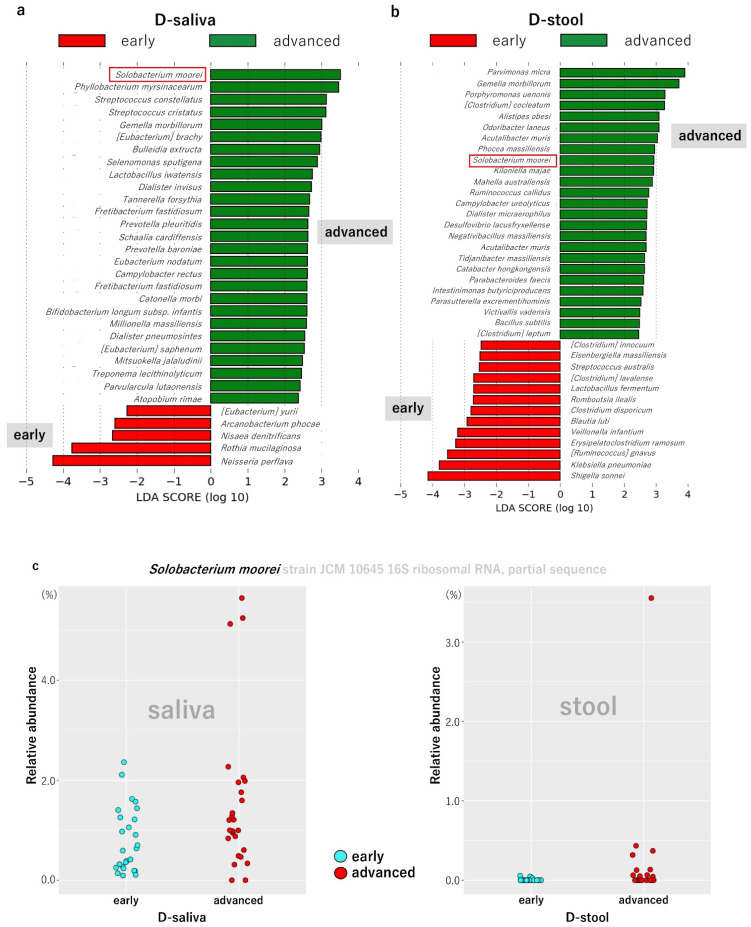
Linear discriminant analysis effect size (LEfSe) comparing bacterial group abundances between “early-stage disêse” (Stage Ⅰ, Ⅱ) and “advanced-stage disease” (Stage Ⅲ, Ⅳ) in (**a**) saliva of patients (D-saliva) and (**b**) stool of patients (D-stool). (**c**) The relative abundance of *Solobacterium moorei* in D-saliva and D-stool was statistically significant in patients with advanced-stage CRC compared to that in patients with early-stage CRC.

**Table 1 cancers-13-03332-t001:** Clinical characteristics of subjects in each group.

Characteristics		Controls (Group N)	CRC (Group D)	*p* Value
Samples (*n*)		51	52	
Gender	Male	26 (51.0%)	33 (63.5%)	0.2004
	Female	25 (49.0%)	19 (36.5%)	
Age (mean±SD)		54.49 (±10.6)	68.52 (±10.6)	<0.01 **
Medical history	Hypertension	12 (23.5%)	23 (44.2%)	0.0256 *
	Diabetes	4 (7.8%)	9 (17.3%)	0.1481
Teeth	Average number of teeth	24.92 (±9.3)	17.7 (±5.2)	<0.01 **
	No decayed teeth	43 (84.3%)	33 (63.5%)	<0.01 **
	With decayed teeth	6 (11.8%)	18 (34.6%)	
Denture	None	47 (92.2%)	27 (51.9%)	<0.01 **
	Using	4 (7.8%)	25 (48.1%)	
Gingival plaque	<1/3 of tooth surface	30 (58.8%)	7 (13.5%)	<0.01 **
	≥1/3 of tooth surface	18 (35.3%)	41 (78.8%)	
Alcohol	None	21 (41.2%)	32 (61.5%)	0.0662
	≤3 days/week	12 (23.5%)	5 (9.6%)	
	≥4 days/week	18 (35.3%)	15 (28.8%)	
Smoking	Never	24 (47.1%)	24 (46.2%)	0.5875
	Experienced	19 (37.3%)	24 (46.2%)	
	Current	8 (15.7%)	4 (7.7%)	
Number of teeth brushing (/day)	≤2	14 (27.5%)	33 (63.5%)	<0.01 **
	≥3	37 (72.5%)	18 (34.6%)	
Dental examination in 3M	None	36 (70.6%)	35 (67.3%)	0.8296
	Yes	15 (29.4%)	16 (30.8%)	

*p* values were calculated using the *x*^2^ test or Welch’s *t* test. * *p* < 0.05, ** *p* < 0.01.

**Table 2 cancers-13-03332-t002:** Clinicopathological characteristics of CRC patients.

Characteristics		*n*	(%)
Region	Cecum	7	13.5
	Ascending colon	7	13.5
	Transverse colon	2	3.8
	Descending colon	5	9.6
	Sigmoid colon	7	13.5
	Rectum	24	46.2
T	T1	7	13.5
	T2≤	45	86.5
N	N0	29	55.8
	N1≤	23	44.2
M	M0	43	82.7
	M1	9	17.3
Stage	Early (Ⅰ, Ⅱ)	26	50.0
	Advanced (Ⅲ, Ⅳ)	26	50.0
Treatment	Surgery	34	65.4
	Chemotherapy	4	7.7
	Neoadjuvant chemotherapy + surgery	12	23.1
	None	2	3.8

## Data Availability

Microbiome analysis (bacterial 16S rRNA gene meta-sequence) data that support the findings of this study have been deposited in the DNA Data Bank of Japan (DDBJ) with the accession codes DRA012322 (https://www.ddbj.nig.ac.jp (accessed on 18 May 2021)).

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
