# Peer review of "Colorectal Cancer Patients Have Four Specific Bacterial Species in Oral and Gut Microbiota in Common—A Metagenomic Comparison with Healthy Subjects"

_cancers, 2021, doi:10.3390/cancers13133332_

Round 1
Reviewer 1 Report
In this study, the authors investigated the paired oral and gut samples from CRC patients and a group of healthy controls. 16S rRNA amplicon sequencing approach was used to profile the microbiome communities of these oral and gut samples. The analyses indicated several oral species are at higher abundance in both saliva and gut samples in the CRC patients. The sample size is not big, but is not very small either, with about 50 subjects in each group. Only 16S rRNA sequencing was used, there was no whole genome metagenomic sequencing. So, the study is only able to find CRC-associated bacteria at genus/species level.
The study design, experimental procedure, and data analyses are well described. All the approaches in data analysis are quite standard. The paper is easy to follow. However, the overall significance of the findings is limited, with only several potential species that could be associated with CRC and the stage of CRC.
Raw 16S sequences are not available from the public repository.
The raw taxonomy abundance data are not available.
Line 124-125: For taxonomic analysis, each OTU was aligned with the Greengene (gg_13_8) (http://greengenes.lbl.gov/)99 reference database.
What is the number "99"?
More details are needed about how the OTUs are aligned with the reference.
Line 272-275:
"F. nucleatum subsp. nucleatum was detected in the saliva of 11/52 (21.15%) patients with CRC, but not detected in the stool of these patients. In contrast, F. nucleatum subsp. vincentii was detected in the saliva samples of 35/52 (67.3%) patients with CRC, as well as in the stool of 6 of these patients."
Here, F. nucleatum was discussed at strain level. We understand that 16S rRNA sequencing often can only identify bacteria at genus or species level. Although strain level identification sometimes is possible with 16S sequencing for some species, more evidence is needed here to support the identification of these F. nucleatum strains.
Author Response
Response to reviewer 1 comments
We thank the reviewer for the important and insightful suggestions provided on the manuscript.
Accordingly, we have revised the manuscript, and point-by-point answers to the reviewer’s comments are shown below.
- The study design, experimental procedure, and data analyses are well described. All the approaches in data analysis are quite standard. The paper is easy to follow. However, the overall significance of the findings is limited, with only several potential species that could be associated with CRC and the stage of CRC.
Response: We thank the reviewer for the comment. We agree with the point raised.
In this study, we identified four bacterial species that commonly exist in oral and gut microbiota of colorectal cancer patients. These species potentially affect the carcinogenesis and progression of CRC, and thus, may represent useful biomarkers for diagnosing CRC.
Nevertheless, we clearly stated that their diagnostic potential is still to be resolved in a future study. Moreover, as addressed in the Discussion section, we believe that whole genome shotgun sequencing should be performed to obtain information on the entire bacterial genome, and additional functional analysis on these bacterial species should be conducted to further explore their role in CRC carcinogenesis and progression. For the biomarker study, we need to establish screening PCR system to confirm the existence of the target bacterial species in saliva, and perform large cohort study.
- Raw 16S sequences are not available from the public repository. The raw taxonomy abundance data are not available.
Response: We submitted the raw 16S sequences and taxonomy abundance data to the DNA Data Bank Japan (DDBJ) database. The public release of the information is still pending (temporary ID: PSUB015203). Once the data are released, we will add the respective accession numbers into the manuscript.
- Line 124-125: For taxonomic analysis, each OTU was aligned with the Greengene (gg_13_8) (http://greengenes.lbl.gov/)99 reference database.
What is the number "99"? More details are needed about how the OTUs are aligned with the reference.
Response: We apologize about this. The number “99” was a typographical error, which we have corrected. We described in more detail the taxonomic analysis.
- Line 272-275: "F. nucleatum subsp. nucleatum was detected in the saliva of 11/52 (21.15%) patients with CRC, but not detected in the stool of these patients. In contrast, F. nucleatum subsp. vincentii was detected in the saliva samples of 35/52 (67.3%) patients with CRC, as well as in the stool of 6 of these patients."
Here, F. nucleatum was discussed at strain level. We understand that 16S rRNA sequencing often can only identify bacteria at genus or species level. Although strain level identification sometimes is possible with 16S sequencing for some species, more evidence is needed here to support the identification of these F. nucleatum strains.
Response: BLAST search for each OTU revealed similarity of 98.563% for F. nucleatum subsp. nucleatum and 99.424% for F. nucleatum subsp. vincentii. We added these data in the revised manuscript.
Reviewer 2 Report
Dear Authors,
Congratulations of the subject selection. However you could do more to improve the paper quality.
- Sample size calculation is missing.
- How did you solve compostionality problem?
- It is important to include the following studies in introduction and discussion:
Chattopadhyay I et al. Exploring the Role of Gut Microbiome in Colon Cancer. Appl Biochem Biotechnol. 2021 Jan 25. doi: 10.1007/s12010-021-03498-9.
Liu W et al. Study of the Relationship between Microbiome and Colorectal Cancer Susceptibility Using 16SrRNA Sequencing. Biomed Res Int.;2020:7828392. doi: 10.1155/2020/7828392.
Zou S, Fang L, Lee MH. Dysbiosis of gut microbiota in promoting the development of colorectal cancer. Gastroenterol Rep (Oxf). 2018;6(1):1-12. doi: 10.1093/gastro/gox031.
Wang Y, Zhang Y, Qian Y, Xie YH, Jiang SS, Kang ZR, Chen YX, Chen ZF, Fang JY. Alterations in the oral and gut microbiome of colorectal cancer patients and association with host clinical factors. Int J Cancer. 2021 Apr 12. doi: 10.1002/ijc.33596. Epub ahead of print. PMID: 33844851.
- The comparative analysis of the raw data from Wang et al. would be very interesting and informative.
- Functional analysis like PICRUSt will be also interesting.
- Conclusions are too speculative. You observed only alterations of oral microbiota in CRC and furher studies are required to conclude more.
Author Response
Response to reviewer 2 comments
We thank the reviewer for the important and insightful suggestions provided on the manuscript.
Accordingly, we have revised the manuscript, and point-by-point answers to the reviewer’s comments are shown below.
Congratulations of the subject selection. However you could do more to improve the paper quality.
- Sample size calculation is missing.
Response: Thank you for the suggestion. As you pointed out, the sample size of our study is small. Nonetheless, the sample size calculation depends on which index and on which bacterial species we employ to detect CRC. Therefore, we could not perform sample size calculation on the study design. We now performed post hoc analysis for sample size using G power application. When we used the relative ratio of each bacterial strain as CRC detecting index, the powers were around 0.85 for Peptostreptococcus stomatis and Solobacterium moorei. So we confirmed that a sample size of 52:51 would be sufficient for these two strains. However, more samples are necessary for another two strains. Hence, we conclude that the sample size used in this study is acceptable as a comparison study.
- How did you solve compostionality problem?
Response: We believe that the problem you pointed out can be solved by LEfSe.
LEfSe first uses the Kruskal-Wallis (KW) sum-rank test and, lastly, applies LDA (linear discriminant analysis). Therefore, this approach ensures that the false positive rate is considerably lower than that obtained by simply performing the KW sum-rank test. Details are given in the reference below, which we cited as the original LEfSe principle:
Segata et al. Metagenomic biomarker discovery and explanation. Genome Biol 2011, 12, R60.
- It is important to include the following studies in introduction and discussion:
Chattopadhyay I et al. Exploring the Role of Gut Microbiome in Colon Cancer. Appl Biochem Biotechnol. 2021 Jan 25. doi: 10.1007/s12010-021-03498-9.
Liu W et al. Study of the Relationship between Microbiome and Colorectal Cancer Susceptibility Using 16SrRNA Sequencing. Biomed Res Int.;2020:7828392. doi: 10.1155/2020/7828392.
Zou S, Fang L, Lee MH. Dysbiosis of gut microbiota in promoting the development of colorectal cancer. Gastroenterol Rep (Oxf). 2018;6(1):1-12. doi: 10.1093/gastro/gox031.
Wang Y, Zhang Y, Qian Y, Xie YH, Jiang SS, Kang ZR, Chen YX, Chen ZF, Fang JY. Alterations in the oral and gut microbiome of colorectal cancer patients and association with host clinical factors. Int J Cancer. 2021 Apr 12. doi: 10.1002/ijc.33596. Epub ahead of print. PMID: 33844851.
Response: Thank you for the valuable advice. We have included these studies in the Discussion section of the revised manuscript.
- The comparative analysis of the raw data from Wang et al. would be very interesting and informative.
Response: Thank you for the insightful advice, and we agree that it would be very interesting. However, as shown in the following literature, human microbiome datasets often change due to differences in sample handling environment or preservation, DNA isolation protocols, and sequencing technologies. Therefore, we are not sure that such analysis would lead to the correct conclusion, even when compared with the raw data of other groups.
Voigt, A.Y.; et al. Temporal and technical variability of human gut metagenomes. Genome Biol 2015, 16, 73.
Sinha, R.; et al. Assessment of variation in microbial community amplicon sequencing by the Microbiome Quality Control (MBQC) project consortium. Nat Biotechnol 2017, 35, 1077–1086.
- Functional analysis like PICRUSt will be also interesting.
Response: Yes, we agree with that. It would be interesting. PICRUSt is more accurate than before, but it is not enough to replace metagenomic analysis. Thus, we decided not to use it in this manuscript.
- Conclusions are too speculative. You observed only alterations of oral microbiota in CRC and further studies are required to conclude more.
Response: As you pointed out, in this study, we identified oral bacterial species commonly found in oral and gut microbiota of CRC patients using the sequence data of a limited region of 16S rRNA. Based on our findings, we suggested that these alterations in bacterial species have some role in the carcinogenesis and progression of CRC and may be useful as biomarkers for CRC. We also believe that further studies are warranted to elucidate the role of these species in CRC. Whole genome shotgun sequencing may provide valuable information on the entire bacterial genome, and functional analysis on these bacterial species should be performed to provide new insights on their contribution for CRC carcinogenesis and progression. For the biomarker study, a screening PCR system still needs to be established to confirm the existence of the target bacterial species in saliva, and a large cohort study should be performed.
We changed the wording in the sentence in the conclusion section regarding these points.
Reviewer 3 Report
In this work, Uchino et al compared the saliva microbiota with the intestinal microbiota of patients with colorectal cancer. They identify 4 bacterial species that can affect the carcinogenesis and progression of colorectal cancer.
The work is very well designed, the methodology is well described and used appropriately. The results support the conclusions reached by the authors. However, both the introduction and the discussion should be extensively revised, as they have some shortcomings.
Major comments:
-The authors dedicate half of the introduction to talking about the incidence of colorectal cancer. It would be enough with 3-4 lines to make it clear the importance of the incidence of colorectal cancer. However, the introduction does not include other research carried out by other authors worldwide on the salivary and intestinal microbiota, which should be included in order to know the state of the art on this subject.
-In Figures 3 and 4, the statistical analysis that compares the two groups of each graph is missing.
-In the discussion, the authors focus on the bacterium F. nucleatum, but do not comment on the role of the four bacteria that appear in saliva and colon of cancer patients: What role do these bacteria play in oral and intestinal physiology, what functions do they have? Are these bacteria pro or anti-inflammatory? Have these bacteria been described in other studies of patients with colorectal cancer?
-In the discussion, the authors describe results on the bacterium F. nucleatum (page 10, lines 271-276) that should go to the results section.
-In the discussion, the authors suggest conducting PCR experiments to confirm the existence of the 4 bacteria. In the opinion of this reviewer, these results should already be included in this work.
-In the discussion, on page 11 (lines 327-336), this paragraph should not be put in the discussion.
-The title of the work should be revised, since the authors do not study the involvement of the oral microbiota in the intestinal microbiota, but rather compare it and obtain the results that there are 4 species in common in the two microbiota.
-The supplementary material is is Japanese.
Minor comments
-Some citations from softwares are missing, such as Qiime2
Author Response
Response to reviewer 3 comments
We thank the reviewer for the important and insightful suggestions provided on the manuscript.
Accordingly, we have revised the manuscript, and point-by-point answers to the reviewer’s comments are shown below.
In this work, Uchino et al compared the saliva microbiota with the intestinal microbiota of patients with colorectal cancer. They identify 4 bacterial species that can affect the carcinogenesis and progression of colorectal cancer.
The work is very well designed, the methodology is well described and used appropriately. The results support the conclusions reached by the authors. However, both the introduction and the discussion should be extensively revised, as they have some shortcomings.
Major comments:
-The authors dedicate half of the introduction to talking about the incidence of colorectal cancer. It would be enough with 3-4 lines to make it clear the importance of the incidence of colorectal cancer. However, the introduction does not include other research carried out by other authors worldwide on the salivary and intestinal microbiota, which should be included in order to know the state of the art on this subject.
Response: Thank you for the advice. We have reduced the description about the incidence of colorectal cancer. To our knowledge, only few research studies on salivary and intestinal microbiota by other authors worldwide have been reported. These references were listed to the manuscript (Lines 60–69).
-In Figures 3 and 4, the statistical analysis that compares the two groups of each graph is missing.
Response: In Figures 3 and 4, we used LEfSe as the statistical analysis method. We put some explanation about this in the respective figure legends.
LEfSe first uses the Kruskal-Wallis (KW) sum-rank test and, lastly, applies LDA (linear discriminant analysis). Therefore, this approach ensures that the false positive rate is considerably lower than that obtained by simply performing the KW sum-rank test. Details are given in the following reference:
Segata et al. Metagenomic biomarker discovery and explanation. Genome Biol 2011, 12, R60.
-In the discussion, the authors focus on the bacterium F. nucleatum, but do not comment on the role of the four bacteria that appear in saliva and colon of cancer patients: What role do these bacteria play in oral and intestinal physiology, what functions do they have? Are these bacteria pro or anti-inflammatory? Have these bacteria been described in other studies of patients with colorectal cancer?
Response: The CRC-related literature of S. moorei and P. stomatis is described in the discussion. S. moorei is also described its physiology and pathological condition. We have added the physiology and relationship with CRC of S. koreensis and S. anginosus. The following statement was added to the revised manuscript (Lines 313–321).
The genus Streptococcus, including S. koreensis and S. anginosus, are also generally considered commensal bacteria of the human oral cavity and can be isolated from the subgingival dental plaque of periodontitis lesions [27,28]. S. anginosus is also recognized as belonging to the normal flora of the human gastrointestinal tract, and there are case reports in which colorectal cancer was found in patients S. anginosus bacteremia [28]. Further research is needed to determine whether S. anginosus infection is a risk factor for CRC or a consequence from cancerous lesion-derived "insult to the normal mucosa allowing pathogens to invade the host circulation.”
-In the discussion, the authors describe results on the bacterium F. nucleatum (page 10, lines 271-276) that should go to the results section.
Response: F. nucleatum is only found in saliva samples of CRC patients. However, this is not a significant result considering our study concept. As previous studies were mainly focused on F. nucleatum, we discussed F. nucleatum based on our LEfSe analysis results described in Figure 3. We examined and considered the presence or absence of F. nucleatum subsp. nucleatum and subsp. vincentii in each patient. It is a consideration rather than a result; therefore, we have included it in this section.
-In the discussion, the authors suggest conducting PCR experiments to confirm the existence of the 4 bacteria. In the opinion of this reviewer, these results should already be included in this work.
Response: Yes, we agree with the reviewer’s opinion. For each OTU obtained as a result of 16S rRNA sequencing, a BLAST search was performed and the name of the species closest to the sequence was retrieved from the candidates. For our future study, we will perform a screening experiment in a large cohort to establish this pattern as a biomarker. For this purpose, we decided to perform a screening PCR with saliva to identify the bacterial species, as it is described as in the discussion.
-In the discussion, on page 11 (lines 327-336), this paragraph should not be put in the discussion.
Response: This paragraph was included in the discussion as a limitation of this study and topics for future research.
-The title of the work should be revised, since the authors do not study the involvement of the oral microbiota in the intestinal microbiota, but rather compare it and obtain the results that there are 4 species in common in the two microbiota.
Response: We agree with the reviewer’s opinion. The title was revised as follows.
“Colorectal cancer patients have four specific bacterial species in oral and gut microbiota in common. – Metagenomic comparison with healthy subjects -”
-The supplementary material is is Japanese.
Response: Since the subjects are Japanese, the informed consent is written in Japanese. We have also submitted an English translation of the ethical code applied in the study.
Minor comments
-Some citations from softwares are missing, such as Qiime2
Response: We agree and have, accordingly, added Qiime2 citation to the manuscript.
Round 2
Reviewer 2 Report
Dear Authors,
The manuscript was significantly improved. Comparison of raw data and PICRUSt are valuable methods used in microbiome analysis. It is a pity that the authors decided not to do it, because it would have increased the scientific value of this publication. However, the analyses carried out are sufficient according to the study aims. I have the following minor comments:
- Explanation concerning sample size calculation should be included in the main text.
- Conclusions: The statements: "These four strains also have the potential to be used as biomarkers in saliva for diagnosing CRC" and " we anticipate that this will lead to the establishment of a risk diagnosis method for CRC using saliva and promote the prevention of CRC through improved oral care" are too speculative and other studies are needed to formulate it. I would recommend to conclude that: “In the present study, we identified alteration of four bacterial species in saliva and gut microbiota with CRC that suggest the possibility of these organisms to have some role in the carcinogenesis and progression of CRC. These four strains also have the potential to be used as biomarkers in saliva for diagnosing CRC. In further studies, it will be necessary to increase the number of samples and perform experiments using PCR analysis”.
Author Response
Response to reviewer 2 comments
We thank the reviewer for the important and insightful suggestions provided on the revised manuscript.
Accordingly, we have revised the manuscript again, and point-by-point answers to the reviewer’s comments are shown below.
The manuscript was significantly improved. Comparison of raw data and PICRUSt are valuable methods used in microbiome analysis. It is a pity that the authors decided not to do it, because it would have increased the scientific value of this publication. However, the analyses carried out are sufficient according to the study aims. I have the following minor comments:
- Explanation concerning sample size calculation should be included in the main text.
Response: Thank you for the advice. The following statement was added to the revised manuscript (Lines 335–340).
Regarding the sample size of this study, we performed post hoc analysis using G power application [33,34]. When we used the relative ratio of each bacterial strain as CRC de-tecting index, the powers were around 0.85 for P. stomatis and S. moorei. So we confirmed that a sample size of 52:51 would be sufficient for these two strains. However, more samples are necessary for another two strains. Hence, we conclude that the sample size used in this study is acceptable as a comparison study.
- Conclusions: The statements: "These four strains also have the potential to be used as biomarkers in saliva for diagnosing CRC" and " we anticipate that this will lead to the establishment of a risk diagnosis method for CRC using saliva and promote the prevention of CRC through improved oral care" are too speculative and other studies are needed to formulate it. I would recommend to conclude that: “In the present study, we identified alteration of four bacterial species in saliva and gut microbiota with CRC that suggest the possibility of these organisms to have some role in the carcinogenesis and progression of CRC. These four strains also have the potential to be used as biomarkers in saliva for diagnosing CRC. In further studies, it will be necessary to increase the number of samples and perform experiments using PCR analysis”.
Response: Thank you for the valuable advice. We have changed the sentence in the conclusion section regarding your recommendation.
Reviewer 3 Report
The authors have met all the points of this reviewer and the paper has improved to be publish now in the present form.
Author Response
Response to reviewer 3 comments
We thank the reviewer for the evaluation provided on the revised manuscript.
The authors have met all the points of this reviewer and the paper has improved to be publish now in the present form.
Response: Thank you very much. Thanks to your important and insightful suggestions, we were able to make a meaningful revision.